# The Emerging Role of LPA as an Oncometabolite

**DOI:** 10.3390/cells13070629

**Published:** 2024-04-04

**Authors:** Theodoros Karalis, George Poulogiannis

**Affiliations:** Signalling and Cancer Metabolism Laboratory, Division of Cancer Biology, The Institute of Cancer Research, 237 Fulham Road, London SW3 6JB, UK; theodoros.karalis@icr.ac.uk

**Keywords:** lysophosphatidic acid, oncometabolite, phospholipid, cancer metabolism

## Abstract

Lysophosphatidic acid (LPA) is a phospholipid that displays potent signalling activities that are regulated in both an autocrine and paracrine manner. It can be found both extra- and intracellularly, where it interacts with different receptors to activate signalling pathways that regulate a plethora of cellular processes, including mitosis, proliferation and migration. LPA metabolism is complex, and its biosynthesis and catabolism are under tight control to ensure proper LPA levels in the body. In cancer patient specimens, LPA levels are frequently higher compared to those of healthy individuals and often correlate with poor responses and more aggressive disease. Accordingly, LPA, through promoting cancer cell migration and invasion, enhances the metastasis and dissemination of tumour cells. In this review, we summarise the role of LPA in the regulation of critical aspects of tumour biology and further discuss the available pre-clinical and clinical evidence regarding the feasibility and efficacy of targeting LPA metabolism for effective anticancer therapy.

## 1. Introduction

Lipids constitute one of the most abundant biomolecules in living organisms, comprising around 50% of the total cell membrane mass. In addition to providing structural support to cells, they are essential intermediates of cellular metabolism and act as potent regulators of both paracrine and autocrine cellular signalling.

Among lipids, lysophospholipids are an important family of complex lipids with potent signalling properties that are classified based on their polar head structure into several subfamilies, including lysophosphatidyl-serine (LPS), lysophosphatidyl-ethanolamine (LPE), lysophosphatidyl-choline (LPC), lysophosphatidyl-inositol (LPI) and lysophosphatidic acid (LPA). Lysophospholipids are found both intracellularly and extracellularly and regulate many cellular functions like migration, invasion and cell cycle progression.

LPA is one of the most important members of the lysophospholipid family, and its role in both human normal physiology and disease has been the subject of extensive research. Specifically, evidence has been accumulating establishing LPA as an emerging oncometabolite, displaying both direct and indirect tumour promoting roles. LPA regulates many different aspects of tumour biology, including metastasis, angiogenesis and the immune system, among others. Although high LPA levels in tumours and plasma fluid from cancer patients have been consistently associated with more aggressive disease, there are many gaps in the knowledge regarding the underlying signalling networks and metabolic rewiring that lead to LPA accumulation. Similarly, the cellular origin of intratumoural and plasma LPA in cancer remains obscure. Therefore, targeting LPA for cancer treatment remains an elusive concept. In this review article, we summarise the latest (<3 years) knowledge and understanding regarding the deregulation of LPA metabolism in human tumours and discuss the potential of targeting this oncometabolite for effective cancer therapy.

## 2. LPA Metabolism

LPA is a glycerophospholipid consisting of a glycerol phosphate backbone in which the sn-1 or sn-2 position is modified with different fatty acid chains. Most sn-2 LPAs are unstable and rapidly undergo acyl-group migration to convert into the more stable sn-1 analogues, which are therefore more abundant. LPA displays a very short half-life of about 3 min [1,2], suggesting that both its synthesis and degradation are tightly regulated. Indeed, there are several enzymes that are capable of producing LPA both intracellularly and extracellularly, and equally numerous are the enzymes that display LPA-catabolising activity. The existence of such a multitude of enzymes controlling LPA synthesis points to the existence of a complex underlying network that is responsible for preserving its systemic homeostasis.

### 2.1. LPA Synthesis

The most studied enzyme in LPA biosynthesis is autotaxin (ATX) or ectonucleotide pyrophosphatase/phosphodiesterase 2 (ENPP2). ATX is an enzyme that is secreted by many different cell types, such as platelets and adipocytes, in the extracellular space, where it can hydrolyse other lysophospholipids to produce LPA (Figure 1). Intriguingly, ATX is the only known enzyme that produces LPA extracellularly. It displays lysophospholipase D activity and can cleave lysophosphatidyl-choline (LPC), lysophosphatidyl-ethanolamine (LPE) and lysophosphatidyl-serine (LPS) to produce LPA [3]. ATX can also hydrolyse nucleotides in vitro, but the significance of this function remains unclear since most of the available studies so far focus on its role in LPA production and signalling. This gap in the literature might be because of the low amounts of nucleotides in the extracellular space, but also because ATX displays 10-fold higher affinity towards its main substrate, LPC, compared to nucleotide substrates [3,4]. Future studies will hopefully investigate this LPA-independent role of ATX in more detail.

Intracellularly, LPA can be derived from many different precursors (Figure 1). Firstly, glycerol-3-phosphate acyltransferases (GPATs) are a family of enzymes that can transfer acyl groups to glycerol-3-phosphate (G3P) using acyl-coA as a donor, leading to the production of LPA. In humans, there are four GPAT isoforms identified so far (GPAT1-4), with GPAT1 and GPAT2 being localised in the mitochondrial outer membrane and GPAT3 and GPAT4 found in the endoplasmic reticulum [5,6]. GPATs transfer acyl groups in the sn-1 position of G3P, leading to the production of sn-1 LPAs [7,8]. Interestingly, AGPAT6, which belongs to the 1-acylglycerol-3-phosphate O-acyltransferase (AGPAT) family, has also been identified to display GPAT activity and therefore contributes to LPA production [9]. This study raises the question of whether there are other, yet undiscovered, proteins with GPAT activity that may contribute to the intracellular pool of LPA.

LPA is also derived from phosphatidic acid (PA) species through enzymatic digestion by phospholipases A (PLA1 or PLA2) (Figure 1). PLA1 specifically removes acyl chains from the sn-1 position of PA, producing sn-2 LPA species, while PLA2 cleaves fatty acid chains from the sn-2 position, leading to the synthesis of sn-1 LPAs [10,11]. While so far only one PLA1 enzyme has been identified in higher eukaryotes, the PLA2 family comprises at least nineteen different mammalian enzymes with phospholipase activity, which are taxonomised in three subfamilies, namely secretory PLA2 (sPLA2), cytosolic PLA2 (cPLA2) and Ca^2+^-independent PLA2 (iPLA2). The sPLA2 subfamily contributes to LPA production by hydrolysing phospholipids from erythrocytes, platelets and whole blood cells that have been exposed to inflammatory stimuli [12]. Of note, the sPLA2-II isoform was the first phospholipase that was shown to display LPA-synthesising capacity from PA [12,13]. Although theoretically both cPLA2 and iPLA2 can produce LPA, evidence regarding their contribution to LPA levels remains scarce [14], and further studies are needed to uncover the role of these phospholipases in LPA biosynthesis.

Arguably, most studies in the current literature are focused on ATX-induced synthesis of extracellular LPA. Given the multitude of enzymes that can potentially contribute to LPA production, our knowledge surrounding the synthesis of intracellular LPA is limited. Furthermore, the absence of potent and specific LPA-detecting agents hinders further studies regarding intracellular LPA. The advent of new technologies and reagents for the detection of LPAs will help to address some of these questions. Last but not least, thus far, it is not known to what extent intracellular LPA synthesis contributes to the amounts of extracellular LPA or even if intracellular LPA can be shed outside the cell (Figure 1). Hopefully, future studies will provide more insights into these intriguing questions.

### 2.2. LPA Catabolism

The enzymes participating in LPA catabolism are equally many, and although the pathways regulating LPA synthesis—and specifically extracellular LPA synthesis—are relatively well understood, their degradation is less well-investigated. The main enzyme responsible for the degradation of extracellular LPA is lipid phosphate phosphatase 1 (LPP1). LPP1 is a magnesium-independent phospholipid phosphatase that is embedded in the plasma membrane and catalyses the dephosphorylation of several extracellular phospholipids, including LPA [2,15,16,17,18,19]. The Km of this enzyme for LPA is ~36 µM, which is much higher than the physiological LPA concentrations and consistent with the fast recycling rates of LPA [2]. The importance of LPP1 for the regulation of extracellular LPA is underscored by the fact that LPP1-null mice not only show enhanced plasma LPA levels but also 4-fold lower degradation of LPA [20]. The other two isoforms of LPPs, LPP2 and LLP3, also contribute to LPA degradation towards mono-acyl-glycerol (MAG), but their role is less well understood [21,22,23]. Interestingly, MAG can be converted to di-acyl-glycerol (DAG) via the action of mono-acyl glycerol acetyl transferase (MGAT), which can then be converted to PA by DAG kinase (DAGK). The fact that PA serves as a substrate for PLAs for LPA production points to the potential existence of an LPA recycling mechanism (Figure 1).

LPA can also be modified by the addition of an acyl unit to its free hydroxyl group by LPA acetyltransferase—also known as acyl-glycerophosphate acyltransferase (AGPAT)—to produce PA (Figure 1). AGPATs are a large family of enzymes, of which the endoplasmic reticulum embedded proteins AGPAT1, 2, 3 and 5 transfer acyl groups to LPAs [24,25,26,27]. The third pathway that contributes to LPA degradation is the hydrolysis of the acyl group to produce G3P by a class of enzymes called lysophospholipases (LPL) (Figure 1). There are two mammalian LPLs that can degrade LPA, namely LPL1 and LPL2. Interestingly, PLAs (PLA1, iPLA2, cPLA2) also display lysophospholipase activity, suggesting that apart from synthesising LPA, they can also degrade it [28,29,30,31,32,33,34]. However, what determines the balance between these opposing reactions and/or whether substrate availability or protein post-translational modifications dictate their function is currently unknown and merits further investigation.

## 3. LPA Receptors

LPA that is either secreted or synthesised extracellularly can interact with LPA receptors (LPARs) in an autocrine or paracrine manner. The presence of LPA in the plasma also suggests that it potentially affects remote anatomical locations, insinuating the existence of hormone-like functions for this phospholipid.

Several LPARs have been identified so far. LPA receptors, including LPAR1, 2 and 3, belong to the endothelial differentiation gene (Edg) family, while LPAR4, 5 and 6 belong to the P2Y purinergic family. LPAR1-6 are all classified as A rhodopsin-like G protein-coupled receptors [35,36]. These receptors can potentially interact with several different G subunits, like Gαi/0, Gs and Gα12/13. The binding of different LPA species to these receptors can activate downstream effector pathways such as Rho GTPase, phospholipase C (PLC), mitogen-activated protein kinase (MAPK) and Ca^2+^, as well as phosphoinositide 3-kinase (PI3K)/protein kinase B (PKB/AKT) signalling, ultimately regulating ample cellular processes such as survival, proliferation, migration and differentiation (Figure 1). Although different LPARs can activate specific signalling pathways, there are also overlapping functions between different receptors. For example, LPA-mediated activation of both LPAR3 and LPAR5 can induce Ca^2+^ signalling [37,38], while LPAR4, 5 and 6 activation can induce Rho GTPase signalling [39,40,41,42]. On the other hand, there is documented evidence that LPARs may antagonise each other. Specifically, in bone development, LPAR1 plays a functional role in promoting osteoblast differentiation, while LPAR4 has an opposing inhibitory effect [43]. While it is yet unclear if there is differential binding affinity of certain LPA species to different LPA receptors, it was recently shown that free and ATX-bound LPAs activate different receptors. Specifically, free extracellular LPA activates only LPAR1, while ATX-bound LPA shows a strong preference for binding to LPAR6 [44]. Following this paradigm, further studies are needed to discern the interaction of specific LPA species with different receptors.

LPA also binds to other G-protein-coupled receptors (GPCRs), including GPPR87 [45], GPPR35 [46] and P2Y10 [47], to induce Ca^2+^ release and subcellular signalling. Moreover, different LPA species interact with non-GPCRs such as the receptor for advanced glycation end products (RAGE) to induce AKT/cyclin D1 signalling [48] and the cation channel transient receptor potential vanilloid 1 (TRPV1)—an ion channel—to promote the release of intracellular Ca^2+^ [49]. Finally, peroxisome proliferator-activated receptor γ (PPARγ) is the only intracellular LPA receptor identified thus far. Intriguingly, LPA-induced PPARγ activation promotes expression of the lipid scavenging receptor cluster of differentiation 36 (CD36) [50], suggesting that CD36-mediated lipid scavenging from the microenvironment could potentially induce LPA production, establishing a positive feedback loop. The existence of an intracellular LPA receptor raises the question of whether LPA-induced PPARγ activation is exclusive to intracellular LPA or can it also be attained by extracellular LPA entering the cell and subsequently activating PPARγ (Figure 1).

## 4. LPA in Human Physiology

LPA displays many crucial roles in human physiology and has a major role in proper development [51], as ATX-knockout mice are embryonic lethal due to neural and vascular defects [52]. Of note, ATX-heterozygous mice that synthesise 50% less LPA display developmental defects associated with vascular and neural system formation [53,54,55]. The effect of ATX-synthesised LPA is partially phenocopied by knocking out LPAR1, 2 and 3, which also results in embryonic lethality [55]. Specifically, loss of LPAR1 causes approximately 50% perinatal lethality, as mice exhibit olfactory bulb and cerebral cortex defects that lead to impaired suckling behaviour [56], whilst LPAR3 deletion results in approximately 30% embryonic lethality due to impaired development of vascular and lymphatic vessels [57].

Moreover, LPA displays crucial roles in spermatogenesis as LPAR1, 2 or 3 knockout mice exhibit reduced mating activity and sperm counts [55], whilst loss of LPAR3 in female mice delays post-fertilisation embryo implantation, which leads to a smaller litter size [58].

LPA can also regulate aspects of the nervous system. Specifically, LPA-mediated activation of LPAR induces RhoA signalling to inhibit neurite extension [59] and regulates neuroblast morphology and neural migration during development [60,61]. Furthermore, LPA-induced signalling regulates several cellular processes of microglia like proliferation and membrane hyperpolarisation [62,63,64,65], whilst in zebrafish, LPA regulates progenitor differentiation to oligodentrocytes [66,67].

Since LPA is a ubiquitously detected molecule, it is not surprising that it influences the properties of many different cell types [51,68]. For example, LPA induces proliferation of brown pre-adipocytes through PI3K, protein kinase C (PKC), Src and extracellular signal-regulated kinase (ERK1/2) pathways [69,70]. Moreover, in naive T-cells, LPA promotes polarisation and motility into the lymph nodes, while in CD4^+^ T-cells, LPA inhibits mitogen-induced migration [71,72,73]. Osteogenesis is also affected by LPA-induced signalling, as LPAR1 promotes osteoblast differentiation while LPAR4 blocks this effect [43]. Intriguing data from another study also showed that LPA/LPAR1 signalling in the bone potentiates osteoclast activity through Ca^2+^/nuclear factor of activated T-cells (NFATc1) signalling that leads to actin cytoskeleton reorganisation [74].

## 5. LPA in Human Cancers

LPA levels in human cancers are highly elevated both in the plasma and the tumour tissues, suggesting that LPA’s role extends beyond a simple biomarker to a crucial mediator of tumour development and progression. Accordingly, here we discuss the roles of LPA in the regulation of several aspects of tumour biology (Table 1 and Figure 2).

### 5.1. LPA in Migration/Invasion/Metastasis

Interest regarding the role of LPA in cancer was first ignited after its discovery as a potential tumour biomarker. Intracellular LPA targets the actin-binding domain of gelsolin, a crucial protein in actin filament formation [75]. The affinity of LPA to gelsolin has been successfully utilised to construct LPA biosensors that detect LPA in ovarian cancer [76]. Given the evidence that high LPA levels are present in tumours, several studies have investigated LPA’s roles in the regulation of migration, invasion and metastases, characterising it as an important oncometabolite across many different tumour types.

#### 5.1.1. LPA in Gynaecological Malignancies

In high-grade serous ovarian cancer, increased LPA levels activate LPAR1 to promote Rho-associated protein kinase (ROCK)- and PKC/ERK-induced myosin phosphatase target subunit 1 (MYPT1) phosphorylation, leading to cytoskeletal changes followed by enhanced migration and entosis [77]. The treatment of ovarian cancer cells with LPA also promotes O-GlcNAcylation of ezrin–radixin–moesin (ERM) cytoskeletal proteins [78], and although the exact mechanism is not entirely understood, LPA-induced activation of ERM suggests the role of LPA in the adhesion and migration of ovarian cancer cells. 

LPAR3, which is overexpressed in ovarian tumours, promotes Gi/MAPKs/nuclear factor kappa-light-chain-enhancer of activated B cells (NF-κβ) signalling to drive tumour growth and metastasis [79]. Notably, LPA activates PI3K/Akt/mTOR/hypoxia-inducible factor 1α (HIF1α) signalling that drives discoidin domain receptor 2 (DDR2) expression to subsequently promote ovarian tumour cell invasion [80]. On the other hand, enhancer of zeste homolog 2 (EZH2) promotes LPAR1 expression by inhibiting micro RNA-139 (miR-139) expression—an LPAR1 targeting miRNA—through inducing H3K27me3 methylation [81], whilst overexpression of miR-367 downregulates LPAR1 levels to inhibit ovarian cancer cell proliferation, invasion and tumour-induced angiogenesis [82]. The ability of LPA to promote invasion could be at least in part driven by its role in altering the cell surface organisation, enhancing, as a result, the assembly of microvilli [83]—a critical step in the formation of metastasis.

LPA also induces the expression of inflammatory cytokines like IL-6/-8 and TNF-α to stimulate breast cancer cell aggressiveness [84]. Furthermore, LPAR1 heterodimerises with C-X-C chemokine receptor type 4 (CXCR-4) and inhibits CXCL12-induced CXCR4 signalling, while LPA/LPAR1 blocks CXCL12/CXCR4-induced migration of triple-negative breast cancer cells [85]. Interestingly, decreased ATX gene expression correlates with promoter methylation, which can also be detected in liquid biopsies from breast cancer patients [86]. Moreover, differential ATX promoter hypermethylation has been noted across many different malignant specimens obtained from hepatocellular carcinoma, melanoma and colorectal cancer compared to their corresponding benign controls [87], suggesting that this regulation might be important for tumour development and progression. Analyses of publicly available data on breast cancer have also shown that LPAR1, 4 and 6 gene expression negatively correlate with tumour aggressiveness, while on the contrary, high LPAR2 expression is associated with increased tumour grade and reduced patient survival [88]. LPAR6 appears to be positively regulated by miR-27a-3p, as knocking down this miRNA induces LPAR6 overexpression to attenuate breast cancer cell proliferation [89].

#### 5.1.2. LPA in Other Tumour Types

Activation of LPAR1 induces Janus kinase 2 (JAK2) and signal transducer and activator of transcription 3 (STAT3) signalling to promote lung cancer cell migration [90]. There is, of course, the possibility that PA can be converted to LPA that subsequently interacts with LPAR1, but this remains to be examined. Furthermore, LPA activates LPAR1, which leads to the stabilisation of the CMTM8 protein, which subsequently stimulates β-catenin signalling to promote pancreatic cell migration and invasion [91]. LPAR5 also interacts with cannabinoid receptor 2 (CB2) to activate AKT signalling and subsequently induce migration and proliferation [92]. Hypoxic colon cancer cells activate HIF2α, which binds hypoxia response elements (HREs) in the ATX promoter to induce the upregulation of ATX expression by histone H3 crotonylation and promote cell migration [93]. Non-alcoholic steato-hepatitis (NASH) is a condition that can promote the development of liver tumours. Interestingly, GPAT1 deletion protects mice from developing NASH [94], suggesting that GPAT1 might potentiate tumour-initiating activities in liver cancer. Indeed, mice with NASH give rise to tumours that are characterised by enhanced levels of LPA, which in turn activates LPAR to promote tumour growth [95]. Brain and muscle Arnt-like protein 1 (BMAL1), a protein involved in circadian rhythm regulation, cooperates with EZH22 to repress the expression of glycerol-3-phosphate acyltransferase mitochondrial (GPAM) and subsequently reduce LPA levels, providing the potential to open a new therapeutic avenue for hepatocellular carcinoma treatment [96]. Nodular tumours of the liver are induced by LPA signalling, and this could be driven, at least in part, by LPA binding to LPAR2, which activates p38 signalling to promote abnormal leptin expression and liver carcinogenesis [97].

In renal cancer cells, LPAR2 activates MAPK/NF-κβ signalling to induce cytokine expression and promote tumour growth and metastasis [98]. In melanoma, the ATX/LPA axis seems to be important for the formation of lung metastasis, as the knockout of ATX markedly reduces this process [99]. In osteosarcoma, platelet-derived LPA binds LPAR1 on the surface of cancer cells to enhance lung metastasis [100].

In thyroid cancer cells, the trinucleotide repeat-containing adaptor 6C-antisense 1 (*TNRC6C-AS1*) regulates the expression of LPAR5. Mechanistically, TNRC6C-AS1 absorbs miR-513c-5p, which can bind and regulate LPAR5 mRNA expression [101]. Furthermore, LPAR5 promotes the proliferation and migration of thyroid cancer cells by activating the PI3K p110β subunit and subsequently AKT/mTOR/S6K1 signalling [102]. PI3K/AKT signalling seems to be at the epicentre of LPA pro-tumourigenic actions, as it has also been suggested to drive LPA-induced proliferation and migration in oesophageal squamous cell carcinoma [103]. LPP1/3 expression is downregulated in oral cancer, whereas ATX displays high expression, and ATX-synthesised LPA subsequently induces cyclooxygenase-2 (*COX-2*) mRNA expression to promote cell migration [104].

In glioma, methylation of ATX mRNA at the 3′-UTR by NOP2/Sun RNA methyltransferase 2 (NSun2) enhances ATX mRNA nuclear-to-cytoplasm transportation and upregulates its expression, a sequence of events that leads to enhanced migratory capacity in glioma cells [105]. Furthermore, glioblastoma-secreted factors induce ATX expression by glial cells and vice versa [106], probably establishing a positive feedback loop to drive high LPA levels in gliomas, which correlate with worse patient survival.

### 5.2. LPA in Proliferation

LPA is also critically involved in the regulation of cell cycles and proliferation. LPA, via LPAR1 and 3, activates the epidermal growth factor receptor (EGFR)/PI3K/mammalian target of rapamycin (mTOR) and Aurora kinase pathways to promote geminin expression and stabilisation, which subsequently regulate DNA replication, cell cycle progression and proliferation of ovarian cancer cells [107]. A similar mechanism of LPA/geminin-controlled cell cycle progression has also been described in gastric cancer cells [108]. Of note, the induction of PI3K/AKT/mTOR/S6K1 signalling by LPA/LPAR5 interactions and concomitant increase in cell proliferation have also been noted in the cases of thyroid and oesophageal squamous cell carcinoma, as described previously {Zhao, 2021 #148} {Liu, 2021 #149}. Non-small-cell lung carcinomas also display high expression of LPAR5, which in turn positively regulates myeloid/lymphoid or mixed-lineage leukaemia, translocated to 11 (MLT11), to stimulate proliferation and migration in vitro as well as tumour growth in vivo [109].

In pancreatic cancer, both ATX and LPA can serve as diagnostic biomarkers for early disease detection [110]. Lipase member H (LIPH)—a PLA1α class enzyme—produces LPA, which activates LPAR/PI3K/AKT/HIF1α signalling to promote pancreatic adenocarcinoma (PDAC) cell colony formation and proliferation. Moreover, high expression of LIPH in orthotopic and patient-derived xenograft (PDX) tumour models is a biomarker of better response to gemcitabine/Ki16425/aldometanib treatments [111].

### 5.3. LPA in Anti-Tumour Immunity

Intratumoural LPA can regulate the immune system to dampen anti-tumour immunity and promote tumour growth and metastasis. In the context of ovarian cancer, LPA induces monocyte differentiation into tumour-associated macrophages (TAMs) that can potentially assist tumour aggressiveness [112,113]. In agreement with the role of LPA as a crucial immunosuppressing agent in the tumour microenvironment, LPA triggers prostaglandin E2 (PGE2) via dendritic cells, which suppresses interferon (IFN) signalling via autocrine prostaglandin EP4 engagement. Moreover, it was identified that LPA-derived immune signatures correlate with a worse response to poly-(ADP-ribose) polymerase (PARP) inhibitors and anti-programmed cell death protein (PD-1) regimens in patients [114]. Keeping up with the role of LPA and its receptors in immune regulation, LPA controls the expression of a wide range of cytokines, including tumour necrosis factor α (TNFα) and interleukin-1β (IL-1β), IL-6, IL-8 and chemokine (C-X-C motif) ligand 1 (CXCL1) through EGFR transactivation to establish the formation of a pro-tumourigenic inflammatory network in ovarian cancer [115]. 

In non-small-cell lung cancer, both ATX mRNA and LPA levels are significantly higher in anti-PD-1 therapy-resistant tumours. Moreover, ATX positively correlates with inflammatory gene signatures and negatively correlates with intratumoural CD8^+^ T-cell numbers [116]. Moreover, in the same cancer subtype, both ATX mRNA and LPA levels are significantly higher in anti-PD-1 therapy-resistant tumours. Additionally, ATX positively correlates with inflammatory gene signatures and negatively correlates with intratumoural CD8^+^ T-cell numbers [116].

In agreement with this evidence, it was recently shown that in CD8^+^ T-cells, LPA/LPAR5 signalling induces metabolic reprogramming and leads to T-cell exhaustion [117,118]. Furthermore, LPA disrupts the CD8^+^ T-cell synapse with the target cell through activation of RhoA/mDia1 signalling, which impairs inositol 1,4,5-trisphosphate receptor type 1 (IP3R1) localisation to the immune synapse [119]. Interestingly, high LPAR6 levels in lung adenocarcinoma tissues are associated with enhanced immune cell infiltration and better overall survival in patients [120]. Similarly, LPAR6 expression in hepatocellular carcinoma patients correlates with a better prognosis and higher tumour immune cell infiltration [121]. On the other hand, melanoma tumour-derived LPA activates LPAR6 on the surface of cytotoxic CD8^+^ T-cells to suppress tumour infiltration, suggesting that LPA functions as a T-cell repellent [122].

In the case of head and neck squamous cell and kidney renal clear cell carcinomas, LPAR2 expression correlates with immune cell infiltration, providing the potential for a potent prognostic biomarker [123].

Reconciling these emerging data with regards to LPA’s roles in the regulation of anti-tumour immunity, it seems clear that LPA can either promote or dampen immune responses, suggesting that its exact functional role is context-dependent. Future research will hopefully clarify which tumour types are more likely to benefit from interventions targeting LPA metabolism to enhance anti-tumour immunity.

### 5.4. LPA in the Tumour Microenvironment (TME)

Apart from controlling immune cell functions and regulating the TME, LPA can also be produced by stromal cells (Figure 3). In breast tumours, although higher levels of ATX have been detected, single-cell RNA-seq analyses revealed that ATX is mostly expressed by endothelial cells and cancer-associated fibroblasts (CAFs) [124]. Furthermore, in CAFs, LPA production leads to high expression of zinc finger E-box-binding homeobox 1 (Zeb1) through LPAR1 and 3/Gi/Rho signalling to promote expression of AREG, resulting in enhanced cancer invasiveness [125]. In agreement with these data, single-cell RNA seq showed that decreased LPP1/3 and increased LPP2 expression were correlated with higher tumour grade, proliferation and tumour mutational burden. The same study also showed that most tumour LPP1/3 is primarily expressed by endothelial cells and CAFs, while LPP2 is expressed by cancer cells [126].

Another important function of LPA is its role in regulating stemness properties [127,128]. For example, in triple-negative aldehyde dehydrogenase-positive (ALDH^+^) breast cancer cells, LPA/LPAR3 induces Ca^2+^ signalling through the transient receptor potential cation channel subfamily C member 3 (TRPC3) channel to induce cancer stem cell populations [129]. In breast tumours, cancer stem cells that grow near the vasculature establish a local interaction with endothelial cells. Importantly, this relationship is established by an LPA/PDK-1/CD36 signalling axis that supports both endothelial cell arteriolar differentiation and the self-renewal of cancer stem cells [130]. In mammary as well as lung tumours, LPA promotes RAGE/AKT signalling to promote epithelial-to-mesenchymal transition (EMT), tumour progression and angiogenesis [131], suggesting that LPA levels are an important factor in regulating neo-angiogenesis—a critical step in tumour progression. Moreover, co-culture of osteosarcoma with endothelial cells induces ATX expression that subsequently cleaves LPC to LPA, which signals through LPAR2 and 3 to increase cancer cell motility [132].

### 5.5. LPA in Drug and Radiotherapy Resistance

LPA functions also extend to the regulation of chemo- and radiotherapy resistance, suggesting that interfering with LPA metabolism might affect the responses of patients to anticancer treatments. In colon cancer cells, upregulation of LPAR2 and LPAR4 helps to overcome the oxidative stress induced by H_2_O_2_ and anticancer drugs such as 5′-fluoro-uracil, irinotecan and oxaliplatin [133]. Moreover, LPAR2 and LPAR5 mediate resistance of colon cancer cells to 5′-fluoro-uracil [134], whilst in the same tumour type, LPA enhances resistance to EZH2 inhibitors through LPAR2 [135], suggesting that co-targeting of ATX/LPA/LPAR2 and EZH2 might be beneficial for colon cancer treatment.

In pancreatic cancer, both ATX and LPA can serve as diagnostic biomarkers for early disease detection [110]. Lipase member H (LIPH)—a PLA1α class enzyme—produces LPA, which activates LPAR/PI3K/AKT/HIF1α signalling to promote pancreatic adenocarcinoma (PDAC) cell colony formation and proliferation. Moreover, high expression of LIPH in orthotopic and PDX tumour models is a biomarker of better response to gemcitabine/Ki16425/aldometanib treatments [111]. Targeting TGF-β in PDAC leads to the skewing of CAFs towards inflammatory CAFs (iCAFs), which secrete ATX in the tumour stroma. The secreted ATX generates LPA, which in turn activates NF-κβ signalling in tumour cells to promote drug resistance [136]. LPA also induces resistance to cisplatin and X-ray irradiation through LPAR2-induced signalling [137]. Resistance to cisplatin in this tumour type is also potentiated by hypoxia, which increases the expression of both LPAR2 and LPAR3 [138,139]. 

Expression of LPAR1 in prostate tumours is associated with better overall survival for patients, probably owing to increased anti-tumour immunity [140]. On the other hand, circular LPAR3 RNA targets Jupiter microtubule-associated homolog 1 (JPT1) to decrease the radiosensitivity of prostate tumour cells [141]. Similar effects of circular LPAR3 RNA have been noted in ovarian cancer, where this circular RNA enhances cisplatin resistance [142]. In agreement with LPA/LPAR’s roles in chemotherapy resistance, LPAR6 enhances the resistance of hepatocellular cancer cells to sorafenib by switching energy metabolism towards lactic acid fermentation instead of oxidative phosphorylation [143], while blocking LPAR2 activation induces resistance to cisplatin, further highlighting the crucial roles of LPARs in drug resistance [144]. Furthermore, LPA reverses the effect of temsirolimus in renal cancer cells by enhancing lipid droplet formation through activation of the MAPK/S6 kinase (S6K1)/DGAT2 pathway [145].

Of note, there is an intricate relationship between LPA signalling and DNA damage response. In cervical cancer cells, LPA exerts cytoprotective functions against doxorubicin-induced cell death [146], whilst irradiation-induced DNA damage, which induces PARP1-mediated methyltransferase 3 and N6-adenosine-methyltransferase complex catalytic subunit (METTL3) expression, enhances LPAR5 mRNA methylation and stability to potentiate cancer cell radiosensitivity [147].

### 5.6. Other Functions of LPA in Tumours

There is recent emerging evidence that LPA affects other tumour aspects, such as senescence, cell metabolism, autophagy and even cancer-related pain. In order for cancer cells to grow and metastasise, they must overcome ‘isolation stress’. Interestingly, isolation stress triggers the expression of LPAR4 to promote the adaptation of cancer cells to stress and subsequently promote survival [148,149]. The role of LPA in cancer-related pain is showcased in fibrosarcoma, where exosomal LPA through LPARs induces cancer-related pain by sensitising C-fibre nociceptors [150]. LPA also affects senescence, as in hepatocellular carcinoma, LPA activates LPAR1, which interacts with myocardin-related transcription factor A (MRTF-A) and filamin A to trigger actin polymerisation and protect against oncogene-induced senescence [151]. In gastric cancer cells, LPA-induced LPAR2 activation promotes β-catenin nuclear localisation through modulation of glycogen synthase kinase-3β (GSK-3β). Moreover, LPA/LPAR2 interaction promotes ATP production and both glycolysis and oxidative phosphorylation [152]. Furthermore, in gastrointestinal tumours, it has been suggested that the ATX/LPA axis lies under the control of Kang-Ai 1 KAT1/CD82, but further investigations are required to discern this connection [153]. The role of LPA in autophagy so far has been investigated in Ras-transformed cells, where inhibition of LPAR3 downregulation blocks autophagy by inhibiting the fusion of autophagosomes with lysosomes [154]. Given the importance of autophagy in cancer cell survival under stress, it is tempting to speculate that LPA/LPAR3 also regulates autophagy in cancer cells, but this remains to be proven.

## 6. Targeting LPA for Cancer Treatment

Since LPA is involved in the regulation of so many critical aspects of tumour development and progression, significant efforts have been focused on targeting this important phospholipid to combat human cancer (Table 2). Arguably, most of the studies have aimed to inhibit ATX, one of the main enzymes responsible for the high plasma and intratumoural LPA levels in cancer patients. The ATX inhibitor IOA-289 potently reduces circulating LPA levels, resulting in slower tumour growth progression in mice [155]. In pancreatic cancer, inhibiting ATX with IOA-289 and galunisertib restores sensitivity to gemcitabine. Importantly, galunisertib treatment in patients increases ATX levels in plasma, while patients with lower ATX levels display higher progression-free survival [136]. IOA-289 treatment also reduces migration as well as the 2D and 3D growth of gastrointestinal cancer cells [156]. Furthermore, it inhibits breast tumour growth in mice, but, in agreement with most tumoural ATX being produced by CAFs and endothelial cells [124], knocking out ATX in adipocytes does not [157]. Interestingly, IOA-289 is the only LPA metabolism-targeting drug that has been evaluated in clinical trials and has so far shown promising clinical outcomes. First of all, its oral administration was able to lower plasma LPA in patients in a randomised, double-blind, placebo-controlled phase 1 study without causing any side effects (clinical trial ID: NCT05027568) [155], suggesting that inhibition of LPA metabolism might offer a safe option to treat tumours. Moreover, a phase 1b trial in patients with metastatic pancreatic cancer is underway to assess the effect of twice-daily oral administration of IOA-289 (clinical trial ID: NCT05586516). This study is expected to be completed by the end of 2024 and will hopefully provide invaluable information regarding the effectiveness of LPA inhibitors in cancer therapy.

The ATX-specific inhibitor S32826 reduces LPA concentrations in pancreatic cells, inhibits FAK phosphorylation and blocks cell proliferation and migration while inducing apoptosis [158]. Another orally available ATX inhibitor, ONO-8430506, enhances the anti-tumour effects of paclitaxel in breast cancer [159], whilst inhibition of ATX with BrP-LPA or LPAR5 with AS2717638 reduces anti-PD-1 treatment resistance in lung cancer [116]. Blocking ATX with PF-8380 in vivo also reduces the peritoneal dissemination of pancreatic cancer cells and decreases malignant ascites [160].

Another strategy to intervene with LPA signalling in tumours is to block the activation of LPARs. Blocking LPAR1 with Ki-16425 in hepatocellular carcinoma interrupts the LPAR1/MRTF-A/filamin A complex—as discussed above—and leads to oncogene-induced senescence [151]. Ki-16425 is also able to block T-cell lymphoma growth both in vitro and in vivo by inhibiting glycolysis, inducing apoptosis and activating the immune system [161]. In breast cancer, Ki-16425 administration can reverse LPA-induced cytokine production, which might lead to reduced tumour progression and aggressiveness [84]. As another proof of function, Ki-16425 blocks LPA-induced proliferation and migration of ovarian cancer cells [79]. Blocking LPAR1 with the LPAR1 inhibitor BMS-986020 results in reduced growth of oesophageal squamous cell cancer [103]. In gastric cancer cells, inhibition of LPA/LPAR2 interaction with XAV393 blocks LPA-induced β-catenin activation and subsequently reduces ATP production, glycolysis and oxidative phosphorylation, indicating the potential of LPAR2 inhibitors as promising agents [152]. The orally available LPAR1 inhibitor ONO-7300243 also inhibits the formation of lung metastases in mice [100], whilst the LPAR6 inhibitor 9-xanthenyl acetate (XAA) renders hepatocellular carcinoma cells sensitive to sorafenib [143]. Of note, embedding the LPAR1 antagonists AM095 and Ki-16425 in liposomes increases their flexibility and provides a viable therapeutic approach to hinder the progression of metastatic breast cancer [162].

Recently, several other small molecules were shown to intervene with LPA/LPAR signalling in cancer cells. Since LPA/LPAR2 signalling is involved in liver carcinogenesis, it has been postulated that berberine, which antagonises the ATX/LPA/LPAR2/p38/leptin axis, could serve as a preventive agent against the development of liver cancer [97]. In ovarian cancer, acacetin blocks LPA-induced RAGE/PI3K/AKT signalling and inhibits the formation of mesothelial cell-initiated malignancy [163]. Moreover, resveratrol treatment antagonises the LPA-induced EMT, PI3K/AKT and JAK/STAT signalling to reduce ovarian cancer cell migration and growth in 3D. Interestingly, LPA drives platinum resistance, and this phenomenon seems to be reversed by inhibiting the Hedgehog pathways and restoring autophagy with resveratrol treatment [164].

**Table 2 cells-13-00629-t002:** Drugs targeting LPA-associated pathways.

Drug Name	Target	Tumour Type	Effect	Reference
IOA-289	ATX	Pancreatic	Restores gemcitabine and galunisertib sensitivity	[136]
IOA-289	ATX	Breast	Decreases tumour growth	[157]
IOA-289	ATX	Gastrointestinal	Reduces migration and 2D/3D growth	[156]
BrP-LPA	ATX	Non-small-cell lung	Restores anti-PD-1 treatment sensitivity	[116]
S32826	ATX	Pancreatic	Induces apoptosis, reduces proliferation and migration	[158]
AS2717638	LPAR5	Non-small-cell lung	Restores anti-PD-1 treatment sensitivity	[116]
Ki-16425	LPAR1	Hepatocellular carcinoma	Promotes oncogene-induced senescence	[151]
Ki-16425	LPAR3	Ovarian	Inhibits proliferation and migration	[79]
Ki-16425	LPARs	T-cell lymphoma	Induces apoptosis, inhibits glycolysis and activates immune system	[161]
Ki-16425	LPARs	Breast	Reduces inflammatory cytokine expression	[84]
ONO-7300243	LPAR1	Osteosarcoma	Decreases lung metastasis	[100]
BMS-986020	LPAR1	Oesophageal squamous cell carcinoma	Reduces tumour growth	[103]
XAV393	LPAR2	Gastric	Inhibits ATP production, glycolysis and oxidative phosphorylation	[152]
XAA	LPAR6	Hepatocellular carcinoma	Increases sensitivity to sorafenib	[143]
Resverastrol	-	Ovarian	Reduces growth, migration and platinum resistance	[164]
Acacetin	-	Ovarian	Prevents mesothelial cell-initiated malignancy	[163]
Berberine	-	Liver	Antagonises ATX/LPA/LPAR2/p38/leptin axis	[97]

## 7. Conclusions

In conclusion, LPA is involved in almost every aspect of tumour biology and is abundant across many different tumour types, affecting a plethora of cellular processes. The effect of LPAs on critical tumour-related processes such as invasion, growth and metastasis suggests that LPA metabolism emerges as an appealing target for the treatment of cancer. Moreover, numerous data suggesting the involvement of LPA in anti-tumour immunity hint that targeting its metabolism could potentially be used in combination with immunotherapies to enhance immune cell killing. Furthermore, initial data from the first clinical trials of LPA metabolism inhibitors indicate that perturbation of this pathway might provide a safe way to treat cancers, as no major side effects have been observed. Although the documented evidence so far seems promising, major questions remain to be answered, including how LPA metabolism is differentially regulated in different tumour subtypes and whether targeting LPA metabolism can indeed yield positive clinical outcomes in cancer patients. 

It seems very likely that as the surge of interest in this oncometabolite continues to rise, more innovative studies to understand the multifaceted role of LPA in cancer biology will come to the fore. Ultimately, current and future knowledge will be distilled into the development of novel anticancer therapies targeting LPA and its signalling. The so far available data support the feasibility of targeting LPA metabolism, and building upon this knowledge is expected to provide more successful anticancer regimens in the foreseeable future.

## Figures and Tables

**Figure 1 cells-13-00629-f001:**
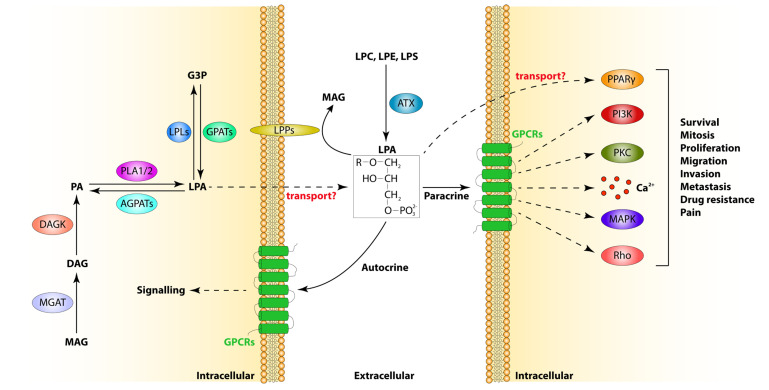
Overview of LPA metabolism and signalling. LPA is synthesised by PLAs and GPATs intracellularly, while extracellular LPA synthesis is catalysed by ATX. LPA is catabolised from LPPs extracellularly and LPLs and AGPATs intracellularly. Whether LPA can be transferred across the plasma membrane remains an open question. Extracellular LPA interacts with LPARs in an autocrine or paracrine manner to promote intracellular signalling like PI3K, PKC, Ca^2+^, MAPK and Rho, which ultimately regulate processes like survival, mitosis, proliferation, migration, invasion, metastasis, drug resistance and pain. There is also one intracellular receptor for LPA, PPARγ. AGPAT: acyl-glycerophosphate acyltransferase; ATX: autotaxin; DAGK: diacyl-glycerol kinase; GPAT: glycerol-3-phosphate acetyl transferase; GPCRs: G protein-coupled receptors; LPL: lysophospholipase; LPP: lipid phosphate phosphatase; PLA1/2: phospholipase A 1/2; MGAT: mono-acyl glycerol acetyl transferase; MAPK: mitogen-activated protein kinase; PI3K: phosphatidyl-inositol-3 kinase; PKC: protein kinase C; PPARγ: peroxisome proliferator-activated receptor γ.

**Figure 2 cells-13-00629-f002:**
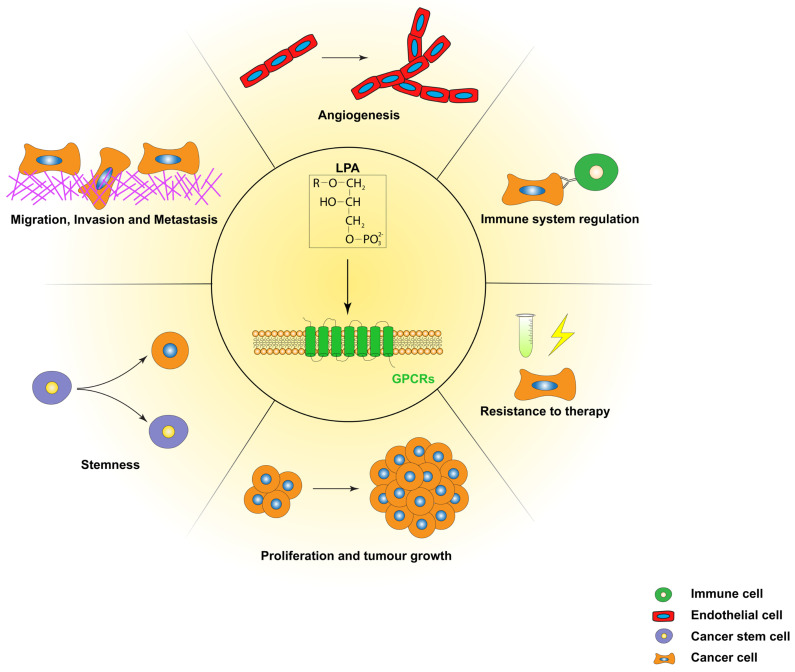
LPA regulates many hallmarks of cancer. LPA acts through cognate G protein-coupled receptors (GPCRs), termed LPA receptors (LPARs), to promote subcellular signalling that ultimately leads to enhanced tumour aggressiveness through induction of processes like angiogenesis, inhibition of anti-tumour immunity, proliferation and tumour growth, cancer stemness, migration, invasion and metastasis.

**Figure 3 cells-13-00629-f003:**
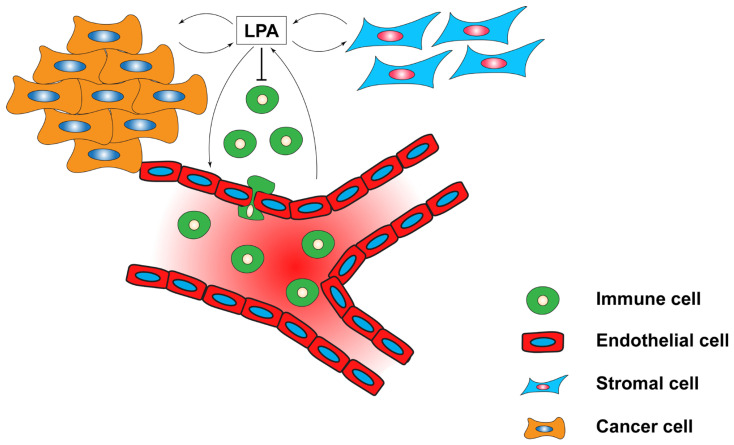
Interplay between LPA and the TME. LPA is not only produced by cancer cells but also by stromal and endothelial cells. Moreover, LPA functions beyond a cell-autonomous manner, and when produced by one cell type, it can also regulate the fate of different cell types in the TME.

**Table 1 cells-13-00629-t001:** Summary of LPA-regulated functions in different tumour types.

Tumour Type	Process
Ovarian	Promotes proliferation, migration, invasion, metastasis, tumour progression, suppresses immune system activation
Breast	Promotes inflammation, cancer stemness, angiogenesis, tumour progression
Lung	Promotes immunotherapy resistance, migration, proliferation, tumour growth, inhibits anti-tumour immunity
Colon	Promotes drug resistance, migration, proliferation
Pancreatic	Promotes proliferation, migration, invasion, tumour growth, inflammatory CAF phenotype, drug resistance
Hepatocellular	Promotes tumour initiation, growth, metastasis, chemotherapy resistance, protects against oncogene-induced senescence
Fibrosarcoma	Promotes drug resistance, pain
Prostate	Promotes resistance to radiotherapy, decreases anti-tumour immunity
Gastric	Reprograms cellular metabolism
Renal	Promotes tumour growth, metastasis, drug resistance
Cervical	Promotes drug and irradiation resistance
Melanoma	Promotes lung metastasis, inhibits anti-tumour immunity
Osteosarcoma	Induces lung metastasis, migration
Thyroid	Promotes proliferation, migration
Oesophageal	Promotes proliferation, migration
Oral	Promotes migration
Glioma	Promotes migration, tumour growth

## Data Availability

Not applicable.

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
