# Peer review of "The Emerging Role of LPA as an Oncometabolite"

_cells, 2024, doi:10.3390/cells13070629_

Round 1

Reviewer 1 Report

Comments and Suggestions for Authors

It is a good review on the role of LPA in cancer.

It will be nice if the authors can discuss as to how the current knowledge on LPA can be extrapolated to the clinic. How do they think that LAP antagonists can be administered to prevent/treat cancer. How much dose, how frequent and route of administration.

Comments on the Quality of English Language

ok

Author Response

We sincerely thank the reviewer for taking the time to read the manuscript and provide this important comment. We have introduced in the text additional information about the assessment of the ATX inhibitor IOA-289 in first clinical trials. It is important to note that so far IOA-289 is the only LPA metabolism-targeting drug that has been tested in clinical trials in the context of an anti-cancer therapy.

Reviewer 2 Report

Comments and Suggestions for Authors

Comments and suggestions to authors

In this article, Karalis and Poulogiannis reviewed the roles of the LPA as oncometabolites. The authors added LPA’s contribution to signaling pathways regulating cellular processes such as mitosis, proliferation, and migration. The investigators detailed that cancer patient tissues show higher LPA levels than healthy tissues, indicating inadequate responses and more aggressive disease. Moreover, LPA boosts cancer cell migration and invasion, promoting metastasis. This review could be a good addition to the literature but needs further revision.

Majors

1.     Please improve the language throughout the manuscript.

2.     Please define the abbreviated terms when they first appear and later use the abbreviations throughout the manuscript.

3.     Please revise the abstract with reference to LPA in cancer therapy.

4.     The introduction is brief; apart from the classification of LPA, please add more details to cover the gaps in LPA roles in cancer biology and therapy.

5.     LPA oncometabolites depend on the tumor microenvironment (TME) since LPA concentration and LPA receptor expression and activity on TME cells considerably affect LPA signaling. The TME may also upregulate LPA synthesis and break down enzymes such as autotaxin (ATX), increasing LPA levels and oncogenic signaling. Roles of LPA specific to the tumor microenvironment are missing from the review, which is critical to understanding the interactions, signaling, and therapy. Please describe it with an illustration.

6.     Section 5 is quite long and complicated in relation to the details based on the random ordering of cancers and cellular processes and pathways. In order to present it better for the journal’s readers, please break down the section into subheadings based on different cancers or specific cellular processes. In fact, I suggest using subheadings based on LPA roles in the different cellular processes because the authors already focused on that in the abstract, so it's also better to present the literature in the same coherent way. For instance, LPA in cancer metastasis, cell invasion, migration, progression and proliferation, and so on.

7.     Likewise, in section 5, please add another subheading specific to the roles of LPA in cancer and immune regulation, and collect all the sentences related to immunology, immune cells, immunosuppression, and detail under this subheading altogether. Also, please illustrate how LPA is involved in immune cells, immune responses and signaling, immunosuppression, and immunotherapy.

8.     In section 6, please add a subheading for the LPA drugs in clinical trials and draw a table on LPA drugs in clinical trials for anti-cancer therapy.

9.     Please briefly revise conclusions specific to the future implications of LPA (and/or its targets) in cancer biology as reviewed, immunology, and how the signaling pathways can be exploited to fill the therapy gaps further.

Minors

1.     L284, Please revise,

Another important function of LPA is its role at regulating stemness properties “to” Another important function of LPA is its role in regulating stemness properties

2.  L291, please revise, tumour progression and angiogenesis in “to” tumour progression, and angiogenesis

3.     L306, please revise, There is of course “to” There is, of course,

4.     L308, please revise, In colon cancer cells upregulation of LPAR2 “to” In colon cancer, cells upregulation of LPAR2

5. L311, please revise, LPA enhance resistance “to” LPA enhances resistance

6.     L372, please revise, In cervical cancer cells LPA exerts “to” In cervical cancer cells, LPA exerts

7.     L375, please revise, In melanoma the ATX/LPA “to” In melanoma, the ATX/LPA

8.      L391, please revise, LPA induced “to” LPA-induced

9.    L403, please revise, In Ras-transformed cells inhibition “to” In Ras-transformed cells, inhibition

10. L443, please revise, ATP production, glycolysis and oxidative phosphorylation “to” ATP production, glycolysis, and oxidative phosphorylation

Comments on the Quality of English Language

1.     L284, Please revise,

Another important function of LPA is its role at regulating stemness properties “to” Another important function of LPA is its role in regulating stemness properties

2.   L291, please revise, tumour progression and angiogenesis in “to” tumour progression, and angiogenesis

3.      L306, please revise, There is of course “to” There is, of course,

4.      L308, please revise, In colon cancer cells upregulation of LPAR2 “to” In colon cancer, cells upregulation of LPAR2

5.  L311, please revise, LPA enhance resistance “to” LPA enhances resistance

6.      L372, please revise, In cervical cancer cells LPA exerts “to” In cervical cancer cells, LPA exerts

7.      L375, please revise, In melanoma the ATX/LPA “to” In melanoma, the ATX/LPA

8.      L391, please revise, LPA induced “to” LPA-induced

9.    L403, please revise, In Ras-transformed cells inhibition “to” In Ras-transformed cells, inhibition

10. L443, please revise, ATP production, glycolysis and oxidative phosphorylation “to” ATP production, glycolysis, and oxidative phosphorylation

Author Response

We sincerely thank the reviewer for devoting the time to read the manuscript and provide these invaluable comments that improved the quality of the text. Accordingly, we present here our point-by-point response.

Majors

  1. 1.     Please improve the language throughout the manuscript.

We have rigorously re-read the manuscript and amended the language. All changes can be found highlighted throughout the text.

  1. Please define the abbreviated terms when they first appear and later use the abbreviations throughout the manuscript.

We have updated the text to define all the terms as they first appear and later used the abbreviations.

  1. Please revise the abstract with reference to LPA in cancer therapy.

We have revised the abstract to include a section referring to drugs targeting LPA metabolism (section 7).

  1. The introduction is brief; apart from the classification of LPA, please add more details to cover the gaps in LPA roles in cancer biology and therapy.

We introduced a section in the introduction discussing the gaps in LPA’s roles in cancer biology and therapy.

  1. LPA oncometabolites depend on the tumor microenvironment (TME) since LPA concentration and LPA receptor expression and activity on TME cells considerably affect LPA signaling. The TME may also upregulate LPA synthesis and break down enzymes such as autotaxin (ATX), increasing LPA levels and oncogenic signaling. Roles of LPA specific to the tumor microenvironment are missing from the review, which is critical to understanding the interactions, signaling, and therapy. Please describe it with an illustration.

We introduced a section about LPA’s roles in TME to fill this gap. We have also included a new figure (Fig. 2) describing the interplay between LPA-cancer cells and the TME.

  1. Section 5 is quite long and complicated in relation to the details based on the random ordering of cancers and cellular processes and pathways. In order to present it better for the journal’s readers, please break down the section into subheadings based on different cancers or specific cellular processes. In fact, I suggest using subheadings based on LPA roles in the different cellular processes because the authors already focused on that in the abstract, so it's also better to present the literature in the same coherent way. For instance, LPA in cancer metastasis, cell invasion, migration, progression and proliferation, and so on. 

We thank the reviewer for this constructive insight. We have re-structured the text to the following sections:

5.1. LPA in migration/invasion/metastasis

5.1.1. LPA in gynaecological malignancies

5.1.2. LPA in other tumour types

5.2. LPA in proliferation

5.3. LPA in anti-tumour immunity

5.4. LPA in the tumour microenvironment (TME)

5.5. LPA in drug and radiotherapy resistance

5.6. Other functions of LPA in tumours

  1. Likewise, in section 5, please add another subheading specific to the roles of LPA in cancer and immune regulation, and collect all the sentences related to immunology, immune cells, immunosuppression, and detail under this subheading altogether. Also, please illustrate how LPA is involved in immune cells, immune responses and signaling, immunosuppression, and immunotherapy.

We have included a section about the roles of LPA in the regulation of tumour immunity (Section 5.3).

  1. In section 6, please add a subheading for the LPA drugs in clinical trials and draw a table on LPA drugs in clinical trials for anti-cancer therapy.

So far, to the best of our knowledge, IOA-289 is the only LPA metabolism-targeting agent that has been investigated in clinical trials with regards to tumour therapy. We have included in the text the relevant information on the two available clinical trials utilizing this drug.

  1. Please briefly revise conclusions specific to the future implications of LPA (and/or its targets) in cancer biology as reviewed, immunology, and how the signalling pathways can be exploited to fill the therapy gaps further. 

We have introduced a small section in the conclusion to include future implications of targeting LPA metabolism as reviewed in the manuscript, and major questions as they emerge from the currently available literature.

Minors

  1. 1.     L284, Please revise,

Another important function of LPA is its role at regulating stemness properties “to” Anotherimportant function of LPA is its role in regulating stemness properties

  1. L291, please revise, tumourprogression and angiogenesis in “to” tumour progression, andangiogenesis
  2. L306, please revise, There isof course “to” There is, of course,
  3. L308, please revise, Incolon cancer cells upregulation of LPAR2 “to” In colon cancer, cellsupregulation of LPAR2
  4. L311, please revise, LPAenhance resistance “to” LPA enhances resistance
  5. L372, please revise, In cervical cancer cells LPA exerts “to” In cervical cancer cells, LPA exerts
  6. L375, please revise, In melanoma the ATX/LPA “to” In melanoma, the ATX/LPA
  7. L391, please revise, LPAinduced “to” LPA-induced
  8. L403, please revise, InRas-transformed cells inhibition “to” In Ras-transformed cells,inhibition
  9. L443, please revise, ATPproduction,glycolysis and oxidative phosphorylation “to” ATPproduction, glycolysis, and oxidative phosphorylation

We thank the reviewer for pointing these details to our attention. Accordingly, we have corrected the phrases as indicated.

Reviewer 3 Report

Comments and Suggestions for Authors

In this manuscript, Karalis and Poulogiannis review the recent literature (mainly original research articles and some reviews) to extract and present the current knowledge on the role of the lysophosphatidic acid (LPA) in human cancer and its potential as a therapeutic target.

The authors start with a short but very informative presentation of the basic biology of this lipid (sections 2 to 4; synthesis, catabolism, receptors of LPA) that is illustrated as well in Figure 1. From this basis, the authors turn to the core and longest part of their review that is dedicated to its specific aspect, that is the involvement of LPA in cancer and the potential of targeting this oncometabolite for cancer therapy (sections 5 and 6).  The potential of LPA as biomarker in various types of cancer and, most importantly, as bioactive lipid and thus as relevant mediator in tumorigenesis is thoroughly analysed and exemplified with data from studies performed in vitro (cell lines), in vivo (animal model) and from clinical studies (human material).  All these data are reported in a synthetic way in Table 1 which offers the reader a convenient summary of the LPA-regulated functions observed in various types of cancer, such as breast, ovarian, lung, pancreatic, liver, brain…just to name a few. The authors continue with a shorter section on the various approaches that are considered and taken in order to try and prevent LPA activities during cancer progression, be it at the level of cancer cells proliferation or migration, or at the level of the regulation of immune cells and other cells in the tumor microenvironment. The list of drugs discussed in this section is presented in Table 2 which, similar to Table 1, is a convenient summary of the types of tumors targeted by each drug and on the biological events that are affected or induced by the drug.

All in all, this rather concise review offers the reader a well-written, well-structured, well-documented and pleasant to read update of the studies performed in the last 3-5 years on the role of LPA in cancer.

I only have a tiny comment, regarding the legend to Figure 1: the authors should list and spell out each abbreviation present in the Figure and the listing should be done in alphabetical order.

Author Response

We sincerely thank the reviewer for their time to read our manuscript and for the kind comments. Accordingly, we have amended the figure legend with the listing presented in alphabetical order.

Round 2

Reviewer 2 Report

Comments and Suggestions for Authors

The authors have addressed the comments and made substantial, satisfactory revisions. The revised manuscript could be considered for potential publication in its present form.